# Evaluation of Clinical and Immunohistochemical Factors Relating to Melanoma Metastasis: Potential Roles of Nestin and Fascin in Melanoma

**DOI:** 10.3390/diagnostics12010219

**Published:** 2022-01-17

**Authors:** Yumiko Yamamoto, Yoshihiro Hayashi, Hideyuki Sakaki, Ichiro Murakami

**Affiliations:** 1Department of Diagnostic Pathology, Kochi University Hospital, Kochi University, 185-1, Kohasu, Oko-cho, Nankoku 783-8505, Japan; ichiro-murakami@kochi-u.ac.jp; 2Equipment of Support Planning Office, Kochi University, 185-1, Kohasu, Oko-cho, Nankoku 783-8505, Japan; jm-hayashiy@kochi-u.ac.jp; 3Department of Pathology, School of Medicine, Kochi University, 185-1, Kohasu, Oko-cho, Nankoku 783-8505, Japan; 4Department of Nutritional Sciences for Well-Being Health, Kansai University of Welfare Sciences, 3-11-1, Asahigaoka, Kahsihara 582-0026, Japan; sakaki@tamateyama.ac.jp

**Keywords:** melanoma, Nestin, Fascin, double immunofluorescence, immunohistochemistry, metastasis, prognosis

## Abstract

For melanoma treatment, an early diagnosis and a complete resection of the primary tumor is essential. In addition, detection of factors that may be related to metastasis is indispensable. A total of 30 Japanese patients with Stage I or II melanoma, diagnosed according to the classification of the American Joint Committee on Cancer, are included in this study. Clinical background (sex, onset age, primary tumor area, existence of remaining cancer cells at the resected tissue margin, and treatment after the primary surgery) and immunohistochemical staining (Nestin and Fascin) on the resected tissue were examined to detect factors statistically related to metastasis. The analysis result has shown that older onset age and positive immunohistochemical expressions of Nestin and Fascin are statistically related to metastasis. To facilitate meticulous observation of Nestin and Fascin expression at different timing (e.g., onset and metastasis), double immunofluorescence staining was performed. Nestin is a class VI intermediate filament protein, initially detected in neural stem cells. Fascin is an actin-bundling protein which regulates cell adhesion, migration and invasion. Nestin and Fascin are suggested to relate to melanoma metastasis, however, the potential role of Fascin is controversial. Analysis of variations in Fascin expression detected in this study may contribute to further investigations concerning potential roles of Fascin for progression of melanoma. This is the first study to report double immunofluorescent staining of Nestin and Fascin in melanoma. Nestin and Fascin double-positive melanoma cells were detected.

## 1. Introduction

Melanoma is classified according to the tumor primary outbreak area as: cutaneous melanoma, mucosal melanoma, uveal melanoma, and unknown-primary melanoma [1,2,3]. Clinicopathologically, melanoma is categorized as Stage I to IV according to the classification of American Joint Committee on Cancer TNM staging system [4]. The National Cancer Center, Kashiwa, Japan 2017 research study showed a moderately favorable five-year survival rate of melanoma of 86.2% (Stage I) and 79.6% (Stage II). However, the survival rate greatly decreased for Stage IV melanoma to 11% [5]. Thus, melanoma is a life-threatening disease when diagnosed at an advanced stage or when metastasis occurs after surgical treatment of the primary tumor [6].

Several studies have been carried out to determine factors that might be useful to predict metastasis of melanoma. A larger tumor size, deeper depth, and formation of ulceration sites are essential factors related to melanoma prognosis as indicated by the TNM staging. Epidemiologically, older age is considered a risk factor, while reports on the effect of patient sex are varied [6]. Several immunohistochemical studies were also completed to examine protein expression in resected melanoma tissue. From these studies, the proteins, Nestin and Fascin, have been suggested to be involved with the process of melanoma metastasis [7,8,9,10,11,12,13,14].

The aim of this study is to determine risk factors to predict metastasis in Japanese patients with Stage I or II melanomas by statistical analysis of clinical data and immunohistochemical staining of Nestin and Fascin. To observe the expression of Nestin and Fascin in each case, at the onset and metastasis, double immunofluorescence staining was performed.

## 2. Materials and Methods

A total of 30 Japanese patients, previously diagnosed with Stage I or II malignant melanoma at Kochi University Hospital, Nankoku, Japan from January 2010, to September 2018, were included in this study. All patients had finished more than three-year follow-ups. Table 1 shows the background of patients. All patients underwent one or more tumor resection surgeries. Tissues obtained during surgery were embedded in paraffin blocks after formalin fixation and preserved. A total of 16 cases developed one or more metastases. During follow-ups, seven patients were transferred to the terminal care unit, and five patients died due to the disease.

## 3. Immunohistochemical Evaluation

In this study, formalin-fixed paraffin-embedded tissue samples were freshly cut into 3 µm thick slices and de-paraffinized. After the extraction of antigen for 30 min at 98 °C in 10 mM citric acid (pH 6.0), the tissue was reacted with anti-Nestin or anti-Fascin antibodies. Following incubation with N-Histofine^®^ and immersion in substrate solution, microscopic photographs were taken with an Olympus digital camera (DP70, Olympus Corporation, Tokyo, Japan). The antibodies and chemical agents used in this study are shown in Table 2.

## 4. Double Immunofluorescence

Antigen was retrieved as mentioned above, and the tissue was reacted with anti-Nestin antibody. After incubation with biotinylated rabbit anti-mouse IgG, the tissue was incubated with FITC-labeled streptavidin solution. After the reaction with anti-Fascin antibody, the tissue was incubated with Texas Red-labeled rabbit anti-mouse IgG, and then counterstained with 4′,6-diamidino-2-phenylindole (DAPI), for fluorescence microscopy, which was carried out using the same camera described above. The antibodies and chemical agents used in the study are shown in Table 2.

## 5. Statistical Analysis

To evaluate immunohistochemical expression of Fascin and Nestin, the Allred scoring system was used. Refs. [15,16,17] Briefly, the proportion of stained cells was categorized as negative (0), <1% (1), 1–10% (2), 11–33% (3), 34–66% (4), and >66% (5) positive. The intensity of the most predominant area was categorized as no (0), weak (1), intermediate (2), or strong (3) staining (Figure 1). Then, an Allred score was provided by adding the proportion and intensity values, which ranged from 0 to 8 points.

Statistical significance to metastasis was examined in relation to the following clinical variables; sex, onset age, primary tumor area, existence of the remaining cancer cells at the resected tissue margin, and treatment after the primary surgery. The relationship between metastasis and immunohistochemical staining of Nestin and Fascin was also evaluated. The relationship to metastasis with sex, primary tumor area, the existence of remaining cancer cells at the resected tissue margin, and treatment after the primary surgery to metastasis was analyzed by Pearson’s correlation coefficient analysis. Wilcoxon signed-rank test was applied to compare metastasis to onset age, and Allred scores of Nestin and Fascin at the primary lesion. Cut-off points for metastases were investigated in the variables that showed a statistical relationship with metastasis. Kaplan–Meier analysis was conducted to investigate the recurrence-free survival distributions between two groups divided by each cut-off point of these variables. Finally, to examine differences of Nestin and Fascin expression values between the primary and metastatic lesions, Wilcoxon signed-rank test was performed in 10 metastasis cases who received the additional surgery. The enumerated analyses were performed for each variable.

All procedures were carried out with the adequate understanding and written consent of each patient. Independent evaluation of immunostaining was performed by two different expert pathologists who were blinded to the clinical data.

## 6. Results

Table 3 shows the results of the statistical analysis between metastasis and each variable. The variables onset age and Allred scores of Nestin and Fascin are significantly related to metastasis. However, the variables sex, primary tumor area, existence of remaining cancer cells at the resected tissue margin, and treatment after the primary surgery were not significantly related to metastasis. Cut-off points for metastasis in onset age, Fascin, and Nestin are also shown in Table 3. Table 4 shows the results of immunohistochemical staining of Nestin and Fascin at the primary and metastatic lesions in 10 cases with metastasis, treated surgically.

Wilcoxon signed-rank test did not detect any statistical differences in Nestin or Fascin expression between primary and metastatic lesions. Figure 2, Figure 3 and Figure 4 show Kaplan–Meier curves and recurrence-free survival distributions between groups differentiated by cut-off points for onset age, Nestin, and Fascin. Table 5 shows detailed primary tumor locations according to the classification of the World Health Organization (WHO), Geneva, categories.

Figure 5, Figure 6 and Figure 7 shows representative images of resected tumor tissue at the primary or metastasis surgery by hematoxylin and eosin staining, Nestin or Fascin Immunohistochemical staining, and double immunofluorescent staining of Nestin and Fascin.

## 7. Discussion

For melanoma treatment, an early diagnosis and complete resection of the primary tumor is essential for a successful prognosis. However, there are cases when a complete resection is difficult. In these cases, pathological observation of the resected tissue taken together with patient clinical background information is crucial to prognose possible recurrence or metastasis. In this study, we attempted to detect risk factors for metastasis after primary surgery of Stage I or II melanoma patients. It was found that advanced patient age and immunohistochemical staining of Nestin and Fascin were statistically related to melanoma metastasis.

Nestin is classified as a class IV intermediate filament protein, which is originally identified in neural stem cells [18,19,20]. Nestin expression is observed in various embryonic cells and tissues, and a correlation of Nestin expression to clinical malignancy has been reported in melanoma and other tumors, such as breast cancer, ovarian cancer, and osteosarcoma [13,14,18,21,22]. Fascin, an actin-bundling protein, plays an important role in the regulation of cell adhesion, migration and invasion [23,24,25,26,27]. Fascin also exists in different tissues of the human body, such as mesenchyme and nervous tissue, however, it is not present in most normal epithelia. Its correlation to malignancy has also been reported in various carcinomas and sarcomas [23,24,25,26,27]. Although, unlike Nestin, there is no commonly accepted role of Fascin regarding melanoma malignancy [7,8,9,10,11]. Thus, there are many studies on Nestin and Fascin, which clarify their histological origins, structures, and roles as useful biomarkers for various tumor malignancy. However, there are few studies regarding the relationship between Nestin and Fascin. The authors have also accomplished several studies on Nestin and Fascin [23,24,28], and, in this study, one of our aims was to reveal the relationship between Nestin and Fascin and the reason why there are contradicting opinions concerning Fascin expression in melanoma malignancy. As shown in Figure 7, tumor cells with Nestin and Fascin double-positive expression were detected in the double immunofluorescence staining of Nestin and Fascin. These cells might be relevant to clinical malignancy of melanoma and also other tumors. Further studies on Nestin and Fascin in melanoma, other cancers and sarcomas with more cases are anticipated.

In this study, Nestin was confirmed to be the most reliable factor to prognose metastasis of melanoma as reported in previous papers. Refs. [13,14] Similarly, a significant relationship between stronger expression of Fascin and metastasis was detected. Patients in our study were those who had been diagnosed with Stage I or II melanoma; cases diagnosed as ‘in situ’ melanoma were not included. In ‘in situ’ melanomas, the epithelial cells occasionally reveal Fascin expression positive, and it is difficult to make a correct judgement for the proportion and intensity in this case, thus, ‘in situ’ melanoma cases were excluded from this study. It is possible that our results may have been affected by the removal of ‘in situ’ melanomas from analysis. Furthermore, there were two cases (Case 13 and 21), where the Fascin Allred score was 8 combined with a successful surgery and metastasis still developed. Other interesting cases included two (Case 16 and 28) where Fascin expression was weak in the primary resected tissue, but it increased at metastasis (Figure 5 and Figure 7). Previous studies report different outcomes related to the role of Fascin regarding melanoma metastasis. These inconsistencies may indicate that additional investigations are needed to determine the potential contribution of Fascin to melanomas. These contributions may be unlike the role of Fascin in other carcinomas and sarcomas, where the role of Fascin is established and generally agreed upon.

Melanomas are classified into two categories as determined by their mutational signatures, anatomic site, and epidemiology; one, etiologically related to sun exposure (sun-exposure melanoma), and the other, resulting not from sun exposure (nonsolar melanoma) [2]. The nonsolar category includes acral melanomas, some melanomas in congenital nevi, mucosal melanomas, uveal melanomas, and others. Acral melanomas are defined as those occurring in the glabrous, that is, non-hair-bearing skin of the volar aspects of the fingers and toes, palms and soles, and nail beds [2]. Table 5 shows the primary locations of the 30 cases included in this study classified according to the WHO. There were 20 nonsolar and 10 sun-exposure melanomas. These data indicate that the nonsolar type of melanoma is dominant in Japanese people as reported in past studies [1]. In this study, the rate of recurrence or metastasis was higher than other reports [5,29]. Several reasons are supposed. First, as we mentioned above, ‘in situ’ melanomas are not included in the study. Secondly, our institution is the only national university hospital in the prefecture, therefore, cases that were difficult to treat at an outpatient clinic, such as patients of extreme high ages, accompanying other serious diseases, and cranial mucosal melanomas, are included. Some cases were performed finger or toe amputation. Among 30 cases of the study, five (Case 17, 20, 21, 22, 25) were pathologically diagnosed as pT4 and eight (Case 3, 14, 15, 18, 23, 26, 27, 29) were diagnosed as pT3. 5/5 cases in pT4 and 6/8 cases in pT3 had recurrence or metastasis. In addition, this study is limited by its small sample size. However, due to the rarity of melanoma in Japan and the rural location of our institution, it was difficult to collect a sufficient number of cases to evaluate the factors relating to malignancy in each melanoma category. In the future, we plan to carry out a multi-institutional study and include a larger number of cases to provide more robust results.

In conclusion, this study confirmed that Nestin is a useful factor to prognose metastasis with Stage I or II melanoma patients after the primary surgery. Follow-ups with intensive care and individualized treatment, including additional chemotherapy or radiotherapy, should be considered for these patients. Stronger Fascin expression was also statistically related to melanoma metastasis. This is the first study that provides double immunofluorescence staining of Nestin and Fascin in melanoma. Nestin and Fascin double-positive tumor cells were detected. Our findings may contribute to further investigations concerning the potential roles of Fascin in melanoma.

## Figures and Tables

**Figure 1 diagnostics-12-00219-f001:**
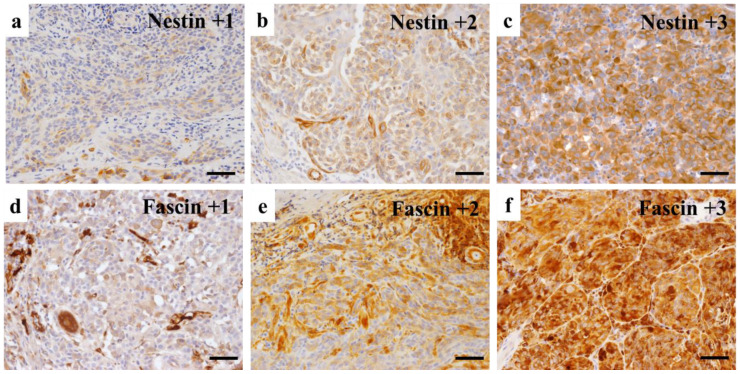
Evaluation of immunohistochemical expression intensity. The intensity of Nestin and Fascin staining was scored as no (0), weak (1), intermediate (2), and strong (3) to convert immunohistochemical expression strength to Allred scoring system. Representative images of the fields are shown for Nestin (**a**–**c**) and Fascin (**d**–**f**) intensity with +1, +2, and +3. (Scale bars: 50 μm).

**Figure 2 diagnostics-12-00219-f002:**
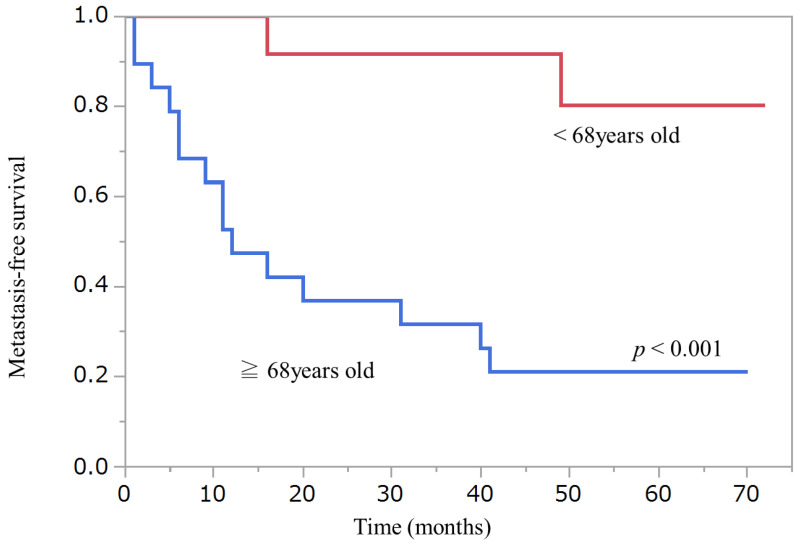
Kaplan–Meier curve and recurrence-free survival distribution between age and metastasis.

**Figure 3 diagnostics-12-00219-f003:**
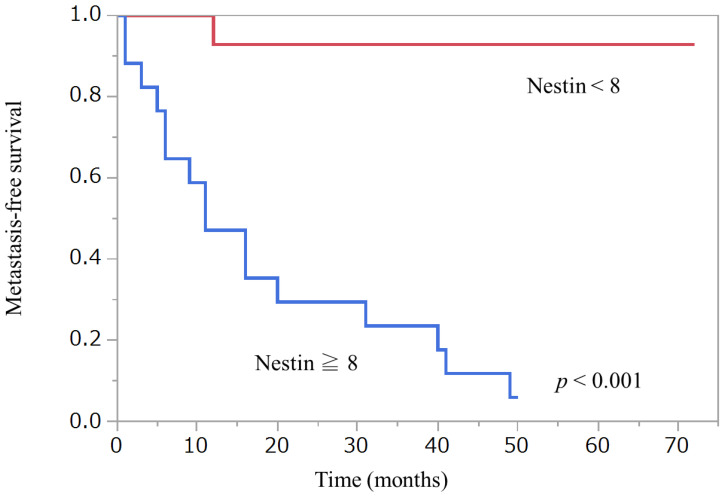
Kaplan–Meier curve and recurrence-free survival distribution between Nestin and matastasis.

**Figure 4 diagnostics-12-00219-f004:**
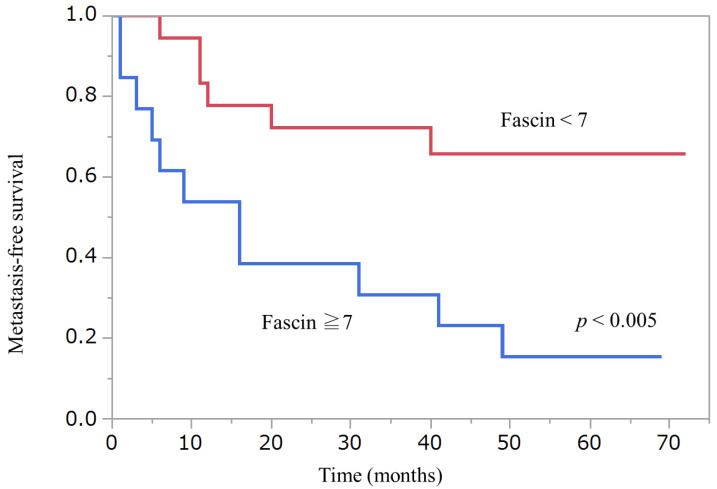
Kaplan–Meier curve and recurrence-free survival distribution between Fascin and metastasis.

**Figure 5 diagnostics-12-00219-f005:**
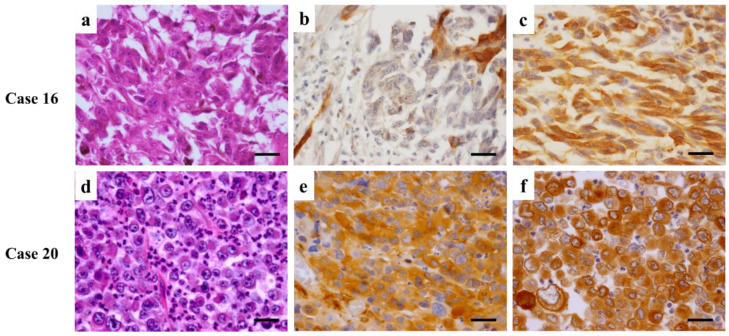
HE and Fascin immunohistochemical staining images of Case 16 and 20. (**a**–**c**) Hematoxylin and eosin staining (**a**) and Fascin immunohistochemical staining (**b**) at the primary tumor, and Fascin immunohistochemical staining (**c**) at metastasis of Case 16. A 76-year-old woman was diagnosed with melanoma on the third finger and received the finger amputation in January 2017. She was followed up regularly without further treatment. In May 2020, metastasis at axillary lymph node was detected and surgical treatment was performed. (**a**,**b**) At the primary lesion, a proliferation of atypical melanocytes with round nuclei is observed. (**b**) The intensity and proportion of Fascin staining are evaluated as +1 and +1, respectively. (**c**) At metastasis, atypical melanocytes with short spindle-shaped nuclei are dominant, and melanin granules are observed in some cells. The intensity and proportion of Fascin staining are evaluated as +3 and +5, respectively. (**d**–**f**) Hematoxylin and eosin staining (**d**) and Fascin immunohistochemical staining (**e**) at the primary tumor, and Fascin immunohistochemical staining (**f**) at metastasis of Case 20. A 77-year-old man had an infection on the thumb in May 2018, and was diagnosed with melanoma in November 2018. The thumb amputation was performed, and no metastasis was detected via positron emission tomography (PET) before surgery. Due to having been treated by chemotherapy for multiple myeloma, systemic chemotherapy for melanoma was not administered. However, in PET of a month later, metastasis at the axillary lymph node was detected, and surgical treatment was performed in February 2019. (**d**,**e**) At the primary lesion, a solid proliferation of tumor cells is observed with the large round nuclei and distinctive nuclear bodies. (**e**) The intensity and proportion of Fascin staining are evaluated as +2 and +5, respectively. (**f**) At the axillary lymph node metastasis, the structure of lymph nodes is unclear, and severe infiltration of tumor cells out of the lymph node can be observed. The intensity and proportion of Fascin staining are evaluated as +3 and +5, respectively. (Scale bars: 50 μm).

**Figure 6 diagnostics-12-00219-f006:**
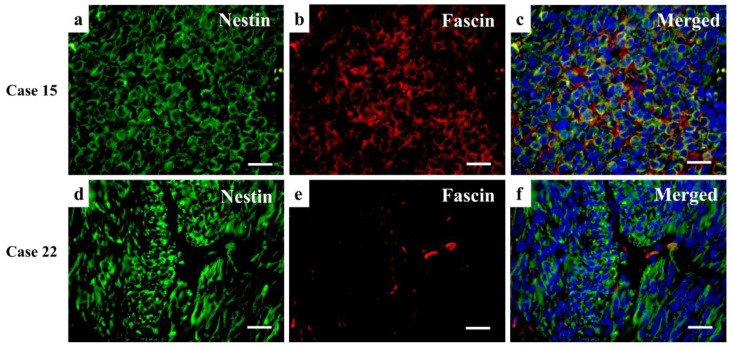
Double immunofluorescence staining images of Case 15 and 22. (**a**–**c**) Immunofluorescence staining of Nestin, Fascin, and the merged staining of the two proteins in the primary tumor of Case 11. A 71-year-old woman was diagnosed with nasal cavity melanoma in September 2015 and had surgical treatment. At the margin of the resected tissue, it was not obvious if tumor cells were removed completely or not, therefore, she had 60 gray radiotherapy. However, in February 2016, distant metastasis was detected. Allred scores of Nestin and Fascin were +8 and +8, respectively, at the primary lesion. Both Nestin and Fascin positive cells densely exist in the tumor. (**d**–**f**) Immunofluorescence staining of Nestin, Fascin, and the merged staining of the two proteins in the primary tumor of Case 22. A 79-year-old man was diagnosed with melanoma on the cheek, and surgical treatment was performed in September 2016. The tumor cells at the margin of the resected tissues were not clearly negative, and topical chemotherapy was planned, which was not undertaken due to patient personal reasons. In May 2018, multiple metastases at the lung and brain were detected. Allred scores of Nestin and Fascin were +8 and 0, respectively, at the primary lesion. Nestin-positive cells are observed at the large area of the tumor, but Fascin is positive only in the vascular endothelial cells. (Scale bars: 50 μm).

**Figure 7 diagnostics-12-00219-f007:**
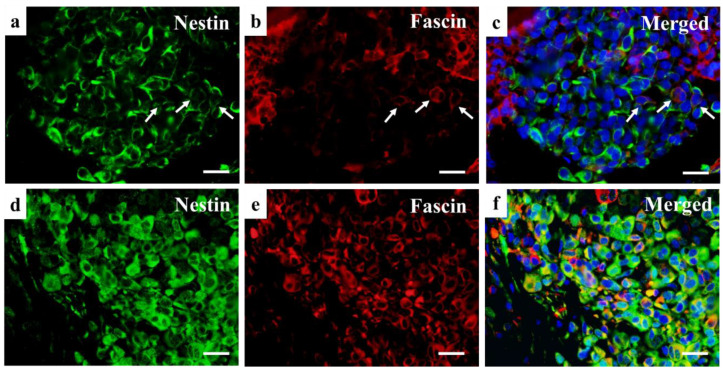
Immunofluorescence staining of Nestin, Fascin, and the merged staining of the two proteins in the primary tumor (**a**–**c**) and metastasis (**d**–**f**) of Case 28. An 88-year-old man was diagnosed with melanoma on the volar aspect of the toe and received amputation in May 2016. No further treatment occurred for the patient and in April 2017, he developed distant metastasis. Allred scores of Nestin and Fascin were +8 and +4 at the primary tumor, and +8 and +8 at metastasis, respectively. At the primary tumor (**a**–**c**), Nestin positive cells are observed in the large area of the tumor, and Fascin is partially positive in the tumor. Nestin and Fascin double-positive cells (arrows) are also observed sporadically in the tumor tissue. At the metastasis (**d**–**f**), both Nestin and Fascin are strongly positive. (Scale bars: 50 μm).

**Table 1 diagnostics-12-00219-t001:** Clinical and immunohistochemical backgrounds of patients.

Case	Age	Sex	Primary Tumor Area	Metastasis	Duration till Metastasis	Remained Tumor Cells	Treatment	Fascin			Nestin		
								Proportion	Intensity	Allred Score	Proportion	Intensity	Allred Score
1	30–39	F	acral cutaneous	-	-	-	C	2	1	3	2	1	3
2		F	mucosal	-	-	+	C	2	1	3	3	1	4
3	40–49	M	acral cutaneous	-	-	RX	C	0	0	0	4	3	7
4		F	non-acral cutaneous	-	-	-	C	1	1	2	5	3	8
5	60–69	M	mucosal	-	-	-	-	5	3	8	4	3	7
6		M	non-acral cutaneous	-	-	-	C	1	2	3	4	2	6
7		F	non-acral cutaneous	+	16	-	C	5	3	8	5	3	8
8		F	acral cutaneous	-	-	-	-	2	1	3	2	3	5
9		F	mucosal	-	-	RX	C	4	2	6	4	2	6
10		M	non-acral cutaneous	-	-	-	C	4	1	5	1	1	2
11		F	mucosal	+	49	+	C, R	5	3	8	5	3	8
12		F	non-acral cutaneous	-	-	-	C	2	1	3	3	2	5
13		F	acral cutaneous	+	3	-	-	5	3	8	5	3	8
14	70–79	F	mucosal	+	12	+	R	5	3	8	5	3	8
15		F	mucosal	+	5	RX	R	5	3	8	5	3	8
16		F	acral cutaneous	+	40	-	-	1	1	2	5	3	8
17		M	non-acral cutaneous	+	6	-	C	5	2	7	5	3	8
18		F	mucosal	+	1	+	C, R	5	3	8	5	3	8
19		M	acral cutaneous	-	-	-	C	1	1	2	2	2	4
20		M	acral cutaneous	+	1	-	- *	5	2	7	5	3	8
21		F	non-acral cutaneous	+	9	-	- *	5	3	8	5	3	8
22		M	non-acral cutaneous	+	20	RX	-	0	0	0	5	3	8
23	80–89	F	mucosal	+	41	+	C, R	5	3	8	5	3	8
24		M	non-acral cutaneous	-	-	RX	C	2	1	3	2	2	4
25		M	non-acral cutaneous	+	6	+	C	1	2	3	5	3	8
26		F	acral cutaneous	-	-	-	-	5	2	7	3	3	6
27		F	acral cutaneous	+	11	RX	C	5	1	6	5	3	8
28		M	acral cutaneous	+	11	-	-	1	3	4	5	3	8
29		F	acral cutaneous	+	16	-	-	5	2	7	5	3	8
30	90–	M	acral cutaneous	-	-	-	-	2	2	4	2	1	3

F: female, M: male, RX: unknown, C: chemotherapy, R: radiotherapy. * These patients had been already taking chemotherapy for other diseases.

**Table 2 diagnostics-12-00219-t002:** Antibodies and chemical agents used in immunohistochemistry.

Antibody	Source	Dilution
Anti-nestin mouse monoclonal antibody	Santa Cruz Biotechnology (Santa Cruz, CA, USA)	1:100
Anti-fascin mouse monoclonal antibody	Dako (Carpinteria, CA, USA)	1:50
Biotinylated rabbit anti-mouse IgG antibody	Dako (Glostrup, Denmark)	1:200
FITC-labeled streptavidin	Dako (Carpinteria, CA, USA)	1:200
Texas Red-labeled anti-mouse IgG antibody	Molecular Probes (Eugene, OR, USA)	1:200
N-Histofine Simple Stain MAX PO (MULTI)	Nichirei Biosciences Inc., (Tokyo, Japan)	ready to use
**Chemical Agent**	**Source**	
4′,6-diamidino-2-phenylindole: DAPI	Sigma (Aldrich, St. Louis, MO, USA)	
3,3′-Diaminobenzidine, tetrahydrochloride: DAB	Sigma (Aldrich, St. Louis, MO, USA)	

**Table 3 diagnostics-12-00219-t003:** Statistical results between each variable and metastasis.

	*p* Value	Cutt-Off Point
Sex	ns	
Onset age	*p* < 0.001	68 years old
Primary tumor area	ns	
Remained cancer cells at the margin	ns	
Treatment	ns	
Fascin at the primary lesion	*p* < 0.005	7 (Allred Score)
Nestin at the primary lesion	*p* < 0.001	8 (Allred Score)

**Table 4 diagnostics-12-00219-t004:** The results of immunohistochemical staining of Nestin and Fascin.

Case	Allred Score of Nestin		Allred Score of Fascin	
	Primary	Metastasis	Primary	Metastasis
7	8	8	8	8
11	8	8	8	8
16	8	8	2	8
18	8	8	8	8
20	8	8	7	8
21	8	4	8	4
23	8	8	8	8
25	8	7	3	0
27	8	2	6	2
28	8	8	4	8
	NS		NS	

**Table 5 diagnostics-12-00219-t005:** Tumor locations classified according to the WHO categories.

Relation with Sun-Exposure	Tissue	Body Area	Rate of Metastasis
sun-exposure melanoma (10) *	non-acral cutaneous (10)	head and neck (3)	2/3
		body trunk (3)	2/3
		limb (4)	1/4
nonsolar melanoma (20)	acral cutaneous (12)	finger (1)	1/1
		toe (7)	3/7
		sole (3)	2/3
		nail head (1)	0/1
	mucosal (8)	nasal cavity (6)	5/6
		conjunctiva (2)	0/2
		Total (30)	16/30

* Numbers in parentheses show the number of patients.

## Data Availability

“MDPI Research Data Policies” at https://www.mdpi.com/ethics (accessed on 20 December 2021).

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
