# Peer review of "Evaluation of Clinical and Immunohistochemical Factors Relating to Melanoma Metastasis: Potential Roles of Nestin and Fascin in Melanoma"

_diagnostics, 2022, doi:10.3390/diagnostics12010219_

Round 1
Reviewer 1 Report
It would be very useful for clinicians to have a way to identify melanoma patients at higher risk of recurrence because there are not affordable biomarkers yet available. In the manuscript, authors describe Nestin and Fascin as possible prognostic factors in stage I and II melanomas, which means patients with early stage disease (no lymphnodal involment) who are 98% cured if stage I and 85% cured if stage II. Unfortunately, authors report a recurrence rate higher than 50% with at least 5 patients who died for their disease: this is much higher than any known statistics. Moreover, chemotherapy and radiotherapy are not the standard treatment in stage I and II melanoma, even in the period of study follow-up. Studies are currently ongoing in order to define the adjuvant role of immunotherapy in stage II, but results are not available yet. So:
1) more info should be given on patients enrolled in the study with each specific histological characteristic of the primary and eventual lymphnode involvement (in this case it would be stage III).
2) more info should be given on treatment received after surgery (usually none): why some patients received chemotherapy and/or radiotherapy? Someone received immunotherapy? Why some patients had received chemo for other diseases and which ones?
3) Are molecular evaluation being performed? How many patients were BRAF/CKIT or NRAS mutated? PDL1 expression? LDH and S100 blood levels?
4) multivariate or univariate statistics?
Minor criticism:
as per journal style, I believe references should be reported like [1-5], not separately as in the manuscript [1][2][3]...last 3 pages should be deleted because blank.
Author Response
Please check my reply to your comments in the attachment. to your comments in the attachment.

Reviewer 2 Report
Authors claim here a prognostic role of nestin and fascin for melanoma, suggesting that both markers might be associated with progressive melanoma. However, the basic assumption here is not correct and must be changed.
1.) Melanoma of stages I and II (skin cancer) does not metasatsize. Metastasis to regional lymph nodes is observed at stage III and a systemic spreading that leads to establishment of metastases at distant site at stage IV. As all of your speciments are from stage I and II melanoma patients, I doubt that these indeed represent metasastes. And therefore no statistical difference between primary and "metastatic" tumors are observed. Therefore the staining for nestin and fascin might not shown differences among metastatic and non-metastatic samples but rather reflect interpatient heterogeneity of samples.
2.) The list comprises many interesting cases of non-sun exposed skin, including mucosal melanoma which is a more aggressive subtype of skin cancer and share common features with acral lentiginous melanoma. However, the nestin and fascin staining might show subtype-specific patterns which has not been adressed as far as I have seen.
3.) Regarding the latter, figure legends must state what is shown. What tumor type (skin, on-skin/acral or mucosal melanoma), what staining, a brief statement. Bars are missing to get an impression of cell size.
4.) Why did authors selected nestin and fascin, what is known about these marker in melanoma and why are these important to investigate? Drivers of stemness, migratory or invasive or proliferative phenotypes?
5.) What antibodies for fascin and nestin have been used?
So, in summary, all these points need to be adressed prior to publication. I would suggest that authors either a.) perform staining of stage IV melanoma or peform an analysis of TCGA data regarding the levels of nesting and fascin in metastatic vs. primary tumors.
Author Response
Please check my reply to your comments in the attachment.

Round 2
Reviewer 2 Report
Dear Editor,
although the manuscript lacks in a functional
validation proving the role of fascin and nestin
such as a knockdown in a certain cell line, I guess th
manuscript is suitable for publication.
Kind regards,
Torben Redmer
Author Response
Reply to the reviewer
Thank you for your precious comments and kind advice to make our paper better.
We have already put several papers concerning Nestin and Fascin in the References and, based on your recommendation, we made a further meticulous search in past papers. However, we could not find any proper paper (literature review) that we may add and refer to the relationship between Nestin and Fascin. This explains the primal background when we had decided to make this study by performing double immunofluorescent staining of Nestin and Fascin on melanoma. No study of simultaneous immunochemical examination of Nestin and Fascin in melanoma has ever been made. According to your advice, we changed the pictures of Figure 7, hence, we can emphasize our novel finding of Nestin and Fascin double positive tumor cells to make our paper more valuable for readers. We do appreciate your advice. We also added description about findings of our immunostainings in the Results and Discussion (blue fonts). Accordingly, we edited figure legends of Figure 7.
*We have made following minor edits, too.
- There was a careless mistake about the case number in the figure 6 legend. Case 15 is correct.
- Reference #28 was added to show our past study in Nestin.
- Careless mistake in the Figure 3 was edited. ‘Nestin≧8’ is correct.
- Commas and periods were input before [ ] to show reference numbers. We corrected them.
- In the Discussion, a sentence ‘Some cases were performed finger or toe amputation.’ was added to indicate severe cases are included in this study.
- In figure legends, to show Allred scores, the order of Nestin and Fascin are changed
- ex) Allred scores of Nestin and Fascin are +8 and 0, respectively
Yumiko Yamamoto, MD, PhD
Kochi University, Kochi, JAPAN
